# Design, Synthesis, and Evaluation of Monoamine Oxidase A Inhibitors–Indocyanine Dyes Conjugates as Targeted Antitumor Agents

**DOI:** 10.3390/molecules24071400

**Published:** 2019-04-10

**Authors:** Xiao-Guang Yang, Yan-Hua Mou, Yong-Jun Wang, Jian Wang, Yan-Yu Li, Rui-Heng Kong, Meng Ding, Dun Wang, Chun Guo

**Affiliations:** 1Key Laboratory of Structure-Based Drug Design & Discovery of Ministry of Education, Shenyang Pharmaceutical University, Shenyang 110016, China; xiaog_yang@163.com (X.-G.Y.); jianwang@syphu.edu.cn (J.W.); 18345384016@163.com (Y.-Y.L.); krh2735@163.com (R.-H.K.); dingmeng74376@126.com (M.D.); 2School of life sciences and biological pharmacy, Shenyang Pharmaceutical University, Shenyang 110016, China; mouyanhua2018@163.com; 3Wuya College of innovation, Shenyang Pharmaceutical University, Shenyang 110016, China; i_maple@163.com

**Keywords:** monoamine oxidase A, prostate cancer, heptamethine cyanine dyes, isoniazid, molecular docking

## Abstract

Monoamine oxidase A (MAOA) is an important mitochondria-bound enzyme that catalyzes the oxidative deamination of monoamine neurotransmitters. Accumulating evidence suggests a significant association of increased MAOA expression and advanced high-grade prostate cancer (PCa) progression and metastasis. Herein, a series of novel conjugates combining the MAOA inhibitor isoniazid (INH) and tumor-targeting near-infrared (NIR) heptamethine cyanine dyes were designed and synthesized. The synthesized compounds **G1**–**G13** were evaluated in vitro for their cytotoxicity against PC-3 cells using the MTT assay, and molecular docking studies were performed. Results showed that most tested compounds exhibited improved antitumor efficacy compared with INH. Moreover, conjugates **G10** and **G11** showed potent anticancer activity with IC_50_ values (0.85 and 0.4 μM respectively) comparable to that of doxorubicin (DOX). This may be attributable to the preferential accumulation of these conjugates in tumor cells. **G10**, **G11**, and **G12** also demonstrated moderate MAOA inhibitory activities. This result and the results of molecular docking studies were consistent with their cytotoxicity activities. Taken together, these data suggest that a combination of the MAOA inhibitor INH with tumor-targeting heptamethine cyanine dyes may prove to be a highly promising tool for the treatment of advanced prostate cancer.

## 1. Introduction 

Global prostate cancer (PCa) incidence and mortality have substantially increased, compounded by an increase in the proportion of the elderly population and in the frequency of diagnosis. Although the majority of cases with metastatic PCa respond to the available therapeutic modalities, hormone-refractory PCa and advanced metastatic PCa remain inevitable [1,2,3,4]. Therefore, novel and highly effective treatments with high tumor-targeting specificity, fewer side effects, and improved efficacy are desirable for hormone-refractory PCa and metastatic PCa.

Monoamine oxidase A (MAOA) is an important mitochondria-bound enzyme that catalyzes the oxidative deamination of monoamine neurotransmitters including serotonin (5-hydroxytryptamine), catecholamines, and biogenic amines [5,6,7,8]. This process subsequently generates their catalytic by-product hydrogen peroxide (H_2_O_2_), a predominant source of reactive oxygen species (ROS), which may predispose cancer cells to DNA damage, leading to tumor initiation and progression. Thus, although MAOA was initially recognized as a crucial neurotransmitter regulator, emerging evidence revealed its significant roles in oncogenesis [9,10,11,12]. Moreover, recent studies have also reported that the overexpression of MAOA could promote the invasion and metastasis of PCa cells by mediating epithelial-mesenchymal transition (EMT) [13,14]. Conversely, the inhibition of MAOA by the inhibitor clorgyline attenuated the proliferation of PCa cells and inhibited tumor growth and metastasis in mouse xenograft models [15,16,17,18,19]. Therefore, MAOA inhibitors can serve as a potentially promising agent for the effective therapy of advanced human PCa.

Near-infrared (NIR) heptamethine carbocyanine dyes are a novel class of heterocyclic polymethine cyanine analogues with cancer imaging ability due to their high extinction coefficients and large Stokes shifts, exhibiting characteristic fluorescence emission in the range of 700–1000 nm [20,21,22]. Recent in vitro and in vivo studies revealed that hypoxia—one of the elements of the tumor endogenous microenvironment—induced the preferential uptake of NIR cyanine dyes by human cancer cells and tissues through the hypoxia inducible factor-1α/organic anion-transporting polypeptides (HIF-1α/OATPs) signaling axis [23,24,25,26,27,28,29,30,31,32]. Second, tumor cells usually possess higher negative mitochondrial transmembrane potentials than normal cells. Therefore, some lipophilic cations preferentially accumulate in the mitochondria of tumor cells [33,34]. The higher mitochondrial membrane potential in tumor cells may play a critical role in the preferential accumulation of heptamethine cyanine dyes in tumors owing to the lipophilic and electropositive characteristics of these dyes [35]. Wang et al. confirmed that NIR heptamethine cyanine dyes IR-780 and MHI-148 could be retained in the mitochondria of tumor cells, increase ROS production, decrease mitochondrial membrane potential, and induce tumor cell apoptosis [36]. Taken together, these findings suggest that heptamethine carbocyanine dyes hold great promise for tumor targeting and to some extent possess antitumor efficacy. 

Isoniazide (INH) is a small molecule inhibitor of MAOA and is now clinically used as the first-line anti-tuberculosis medicine. Studies have shown that INH hydrazone and N-acetylated derivatives are more efficient and less hepatotoxic than INH because of the blockage of the terminal amino group of hydrazine [37,38,39]. Herein, we designed and synthesized a series of novel conjugates (Figure 1) through the conjugation of structurally diverse heptamethine carbocyanine dyes with INH via hydrazine bond and investigated their cytotoxicity against PC-3 cells. Further, molecular docking studies were performed to elucidate the binding mode of these conjugates with MAOA. MAOA inhibition assay was also carried out.

## 2. Results and Discussion

### 2.1. Chemistry

The synthetic route for the type **I** compounds is outlined in Scheme 1. Intermediates **1** and **2** were both synthesized by electrophilic substitution of 5-substitutional 3*H*-indole. The synthesis of key intermediate **4** was achieved according to procedures described in the literature [26]. Briefly, both intermediates **1** and **2** condensed with **3** via base-catalyzed (AcONa) aldol condensation reaction to give intermediate **4**. The target compounds **G1**–**G3** were generated through the coupling reaction of intermediate **4** with INH in the presence of dicyclohexylcarbodiimide (DCC) and dimethylaminopyridine (DMAP). 

As illustrated in Scheme 2, symmetric heptamethine cyanine dye intermediate **6** was synthesized by one-step base-catalyzed (Et_3_N) aldol reaction of intermediates **1** and **2**. However, asymmetric heptamethine cyanine dye intermediate **6** was afforded by the two-step aldol condensation reaction of intermediates **1**, **2**, and **5** in Ac_2_O with pyridine as catalyst. Finally, the desired compounds **G4**–**G12** were generated by the coupling of **6** with INH in anhydrous DCM with DCC and DMAP acting as coupling agents. Meanwhile, the type **III** compound **G13** was obtained in the sequence of steps outlined in Scheme 3.

### 2.2. Cytotoxicity against PC-3

The antitumor activities of **G1**–**G13** were evaluated by MTT (3-(4,5-dimethylthiazol-2-yl)-2,5-diphenyltetrazolium bromide) assay using a human PC-3 cell line. The IC_50_ values of **G1**–**G13** against PC-3 cells are summarized in Table 1.

MTT assay results revealed that nearly all tested compounds showed improved antitumor activity when compared with their parent compound, the MAOA inhibitor INH. The moderate cytotoxicity of these conjugates may be due to their preferential uptake by PC-3 cells mediated by OATPs and subsequent accumulation in the mitochondria of PC-3 cells, disrupting mitochondrial activities. The fact that INH exhibited negligible cytotoxicity may be attributed to its poor passive diffusion into PC-3 cells. **G10**, **G11** (IC_50_ values of 0.85 and 0.40 μM, respectively) possessed potent cytotoxicity comparable to that of doxorubicin (DOX) (IC_50_ = 0.21 μM). **G10** and **G11** have very similar chemical structures except for the different chain length of carboxyalkyl connected to the N atom in indole moieties. Their structural similarity may explain their similar antitumor efficacy. These results indicated that the conjugation with INH was conducive to the enhancement of anti-tumor activity against PC-3 in vitro.

### 2.3. MAOA Inhibitory Activity of G10, G11, and G12

Because the prostate cancer cell line LNCaP expressed a relatively high level of MAOA [17,40], we tested the inhibitory effect of compounds **G10**, **G11,** and **G12** on MAOA using LNCaP cells (Table 2). Our data showed that three synthesized novel compounds displayed an enhanced inhibitory action on MAOA levels as compared with the parent compound INH, while clorgyline—another MAOA inhibitor—exhibited the most potent inhibitory effect. The fact that heptamethine dye–INH conjugates possessed higher MAOA inhibition efficacy than INH may be attributed to the modification of the terminal hydrazine group in INH. This assay result is consistent with that of the cell viability assay, suggesting that the tested compounds inhibited cell growth by suppressing MAOA activity.

### 2.4. In Silico Molecular Docking

To clarify the underlying mechanism of action of these compounds at the molecular level, molecular docking of **2BXR** and **G11** was performed. **2BXR** is a human monoamine oxidase A (hMAOA). De Colibus et al. revealed that the active binding site of hMAOA was formed primarily by a single cavity consisting of residues 210–216 (all active site residues: Tyr69, Gln74, Val91, Val93, Leu97, Ile207, Phe208, Ser209, Val210, Glu216, Cys323, Ile325, Ile335, Leu337, Met350, Phe352, Tyr407, Tyr444) [41]. Since **G11** showed the best in vitro antitumor activity, it was selected as the ligand for this docking analysis. The molecular docking analysis showed the heptamethine chain and indole moieties of **G11** could be integrated closely with the hydrophobic active site cavity (Figure 2) with the low binding energy of (ΔG) −12.27 kcal/mol. The docked pose of **G11** (in 2BXR) was enclosed by a partially hydrophobic cavity with amino acid residues including Tyr69, Val93, Leu97, Ile180, Ile207, Phe208, Phe208, Val210, Cys323, Ile325, Ile335, Leu337, Met350, Phe352, Tyr407, and Tyr444. A hydrogen bond interaction between O44 and Ser209 was also observed. 

## 3. Materials and Methods

### 3.1. Materials

All reagents and solvents were obtained from commercial sources and were used as received unless otherwise stated. Phenylhydrazine hydrochloride, 4-bromophenylhydrazine hydrochloride, 4-methylphenylhydrazine hydrochloride, 4-methoxyphenylhydrazine hydrochloride, 6-bromohexanoic acid, ethyl 4-bromobutyrate, and 1-bromobutane were purchased from Meryer (Shanghai, China). 3-Methyl-2-butanone, 3-methyl-2-pentanone, 1,4-butane sultone, and isoniazid were purchased from Aladdin (Shanghai, China). Clorgyline was obtained from Bide Pharmatech Ltd. (Shanghai, China). All other reagents were synthesized in our laboratory. 1640 DMEM and 0.25% trypsin EDTA were obtained from Gibco (Sigma-Aldrich, St. Louis, MO, USA). (3-(4,5-Dimethylthiazol-2-yl)-2,5-diphenyltetrazolium bromide). MTT reagent was acquired from Sigma-Aldrich (St. Louis, MO, USA). PC-3 and LNCaP cells were obtained from the Cell Bank of the Chinese Academy of Sciences (Shanghai, China).

### 3.2. Molecular Docking

The crystal structure of MAOA with a resolution of 3 Å was retrieved from the Protein Data Bank (PDBID: **2BXR**). AutoDock 4.2 suite, an automated docking tool that uses a Lamarckian genetic algorithm (LGA), was used to perform the molecular docking simulations [42]. Briefly, all the bound water molecules, ligands, and co-factors were eliminated from the protein, and the polar hydrogen was added to the proteins. Gasteiger charges were computed in AutoDockTools 4.2 [43]. 2D structures of **G11** were constructed using ChemBioDraw ultra12.0 and were transformed to 3D. Subsequently, 3D structures of the aforementioned compounds were also deduced and then the structure was energetically minimized by using an MMFF94 force field. 

In the docking process the grid point maps were calculated using AutoGrid 4.2. The program AutoGrid was used to generate the grid maps. Each grid was centered at the structure of the corresponding receptor. The grid dimensions for the grid maps were fixed at a spacing of 0.375 Å with the grid box of size 90 × 70 × 60 (number of points in x-, y-, and z- axes for both proteins). For each docking experiment, default parameters for docking run included number of final conformations, 100; population size, 150; maximum number of energy evaluations, 2.5 × 10^6^, and maximum number of generations, 27,000. Some low-binding-energy docked poses were investigated in detail for their binding interactions with the binding cavity of the target protein, using Discovery Studio Visualizer (DSV). The docked compound complexes were built using the low-free-energy binding positions.

### 3.3. Cytotoxicity Evaluation

The in vitro cytotoxicity activity of **G1**–**G13** were measured by the colorimetric MTT assay. Briefly, PC-3 cells were incubated in 96-well plates at a density of 3 × 10^3^ cells/well for 24 h. Then, the cells were exposed to various concentrations of tested compounds, INH and DOX, respectively for 96 h. After incubation, 10 μL of MTT (5 mg/mL) solution was added and the plates were incubated for 4 h at 37 °C. Next, the medium was replaced with 200 μL of DMSO to dissolve the formed formazan crystals. The absorbance was measured at 570 nm on a microplate reader (Synergy-HT, BioTek Instruments, Winooski, VT, USA).

### 3.4. MAOA Inhibition Activity Assay

In a 10-mm dish, 6 × 10^5^ LNCaP cells were plated in medium supplemented with 10% FBS. After 24 h, cells were treated with various concentrations (10 pM, 1 nM, 100 nM, 10 μM) of the tested compounds **G10**, **G11**, **G12**, isoniazid, and clorgyline for 48 h. The reaction products were extracted and MAOA activities were measured as described in the introduction for the Cell MAOA Assay Kit (Shanghai Chengong Biotechnology, CHN, Lot No: 1-201838-10). The MAOA activity was assessed based on the substrate p-tyramine level after treatment with MAOB inhibitor pargyline.

### 3.5. General Procedure for the Synthesis of Compounds ***G1**–**G3***

(*E*)-2-Chloro-3-(hydroxymethylene)-cyclohex-1-enecarbaldehyde (**3**) (500 mg, 2.9 mmol), 5-methoxyl-1-(5-carboxypentyl)-2,3,3-trimethyl-3*H*-indol-1-ium bromide (**1a**) (2.23 g, 5.8 mmol) and sodium acetate (401 mg, 2.9 mmol) were suspended in acetic anhydride, the mixture was stirred at 60 °C for 1 h. Next the resulting solution was poured into water. Precipitation was allowed to proceed for 6 h and then mixture was filtered to obtain intermediate **4a** as a gold solid. DCC (104 mg, 0.506 mmol) and DMAP (6 mg, 0.051 mmol) were added to a solution of **4a** (400 mg, 0.506 mmol) in DCM in an ice-salt bath. INH (58 mg, 0.422 mmol) was added dropwise, the mixture was stirred for 24 h, and was then filtered. Filtrate was evaporated to dryness under reduced pressure and the residue was purified by silica gel column to yield **G1** (50 mg, 13%). **G2** and **G3** were prepared by the same procedure as **G1** (Appendix A).

*2-(2-(3-(2-(5-Methoxyl-1-(5-carboxypentyl)-3,3-dimethyl-indol-2-ylidene)ethylidene)-2-chloro-1-cyclohexen-1-yl)vinyl)-5-methoxyl-1-(6-(2-isonicotinoylhydrazinyl)-6-oxohexyl)-3,3-dimethyl-3H-indol-1-ium bromide* (**G1**). MS (ESI) *m*/*z*: 862.4 [M + H]^+^. ^1^H NMR (400 MHz, CDCl_3_) δ 8.63–8.56 (br, 2H), 8.14–8.12 (m, 2H), 7.89–7.83 (br, 2H), 7.11 (d, 1H), 6.89 (s, 2H), 6.87 (s, 2H), 6.84 (s, 2H), 6.14 (d, 1H), 6.00 (d, 1H), 5.92 (d, 1H), 4.09–4.01 (br, 2H), 3.94–3.87 (br, 2H), 3.82 (s, 3H), 3.80 (s, 3H), 2.58–2.54 (br, 2H), 2.50–2.45 (br, 2H), 2.43–2.34 (br, 2H), 2.23 (d, 4H), 1.83–1.78 (br, 2H), 1.77–1.70 (br, 6H), 1.65 (s, 6H), 1.63 (s, 6H), 1.56 (d, 4H). ^13^C NMR (151 MHz, CD_3_OD) δ 180.10, 174.51, 172.99, 172.86, 166.31, 159.81, 159.74, 150.86, 149.56, 144.20, 144.16, 144.02, 143.94, 141.91, 141.89, 136.92, 136.90, 126.93, 126.90, 124.92, 124.73, 122.96, 118.49, 114.75, 112.89, 112.82, 109.80, 101.80, 101.67, 66.69, 56.26, 50.57, 50.54, 48.94, 45.15, 45.08, 38.22, 34.22, 28.15, 28.11, 27.89, 27.65, 27.15, 27.12, 27.06, 26.94, 25.88, 23.76, 21.97, 15.26.

*2-(2-(3-(2-(5-Bromo-1-(5-carboxypentyl)-3,3-dimethyl-indol-2-ylidene)ethylidene)-2-chloro-1-cyclohexen-1-yl)vinyl)-5-bromo-1-(6-(2-isonicotinoylhydrazinyl)-6-oxohexyl)-3,3-dimethyl-3H-indol-1-ium bromide* (**G2**). MS (ESI) *m*/*z*: 960 [M + H]^+^. ^1^H NMR (400 MHz, CD_3_OD) δ 8.71 (d, 2H), 8.42 (d, 1H), 8.39 (d, 1H), 7.81 (dd, 2H), 7.71 (d, 2H), 7.69–7.65 (m, 1H), 7.58–7.57 (m, 1H), 7.56 (dd, 1H), 7.55 (d, 1H), 7.28 (d, 1H), 7.27 (d, 1H), 6.30 (d, 1H), 6.28 (d, 1H), 2.75–2.71 (br, 2H), 2.69 (t, 2H), 2.36 (t, 2H), 2.20 (t, 2H), 1.92–1.89 (m, 4H), 1.79 (dt, 2H), 1.73 (d, 12H), 1.71–1.65 (m, 4H), 1.58 (dt, 7.7 Hz, 2H), 1.50–1.44 (m, 4H). ^13^C NMR (151 MHz, DMSO) δ 175.40, 171.45, 163.96, 156.11, 150.48, 147.97, 143.87, 141.01, 140.49, 139.78, 132.07, 130.69, 130.45, 129.95, 129.63, 128.38, 124.89, 124.77, 124.56, 123.74, 121.51, 110.78, 108.44, 108.17, 108.15, 107.96, 99.69, 99.62, 93.24, 48.85, 45.44, 40.14, 33.45, 33.24, 28.71, 28.12, 27.72, 27.43, 27.22, 26.61, 26.51, 26.15, 26.00, 25.47, 25.22, 24.97, 21.27, 19.66.

*2-(2-(3-(2-(5-Methoxyl-1-(5-carboxypentyl)-5-ethyl-3-methyl-indol-2-ylidene)ethylidene)-2-chloro-1-cyclohexen-1-yl)vinyl)-5-methoxyl-1-(6-(2-isonicotinoylhydrazinyl)-6-oxohexyl)-5-ethyl-3-methyl-3H-indol-1-ium bromide* (**G3**). MS (ESI) *m*/*z*: 891.5 [M + H]^+^. ^1^H NMR (400 MHz, CDCl_3_) δ 8.59 (s, 1H), 8.21 (d, 1H), 8.14 (d, 1H), 7.77–7.70 (br, 2H), 7.68–7.62 (br, 1H), 7.57–7.50 (br, 1H), 7.18–7.10 (br, 1H), 7.07 (d, 1H), 6.97 (d, 2H), 6.91–6.87 (m, 1H), 6.85 (d, 1H), 6.16 (d, 2H), 5.98 (d, 1H), 4.06–3.97 (br, 2H), 3.95–3.88 (br, 2H), 3.83 (d, 6H), 2.56–2.50 (br, 2H), 2.50–2.44 (br, 2H), 2.38–2.32 (br, 2H), 2.30–2.21 (br, 4H), 2.13–2.07 (m, 2H), 2.00–1.94 (br, 4H), 1.73–1.69 (m, 4H), 1.64 (d, 6H), 1.53–1.48 (br, 2H), 1.44–1.39 (br, 2H), 1.26 (d, 2H), 1.21 (d, 6H). ^13^C NMR (151 MHz, CD_3_OD) δ 174.66, 171.35, 171.17, 166.41, 159.80, 159.75, 150.87, 143.53, 143.29, 142.20, 142.13, 141.71, 137.99, 137.96, 128.75, 127.04, 126.93, 125.23, 125.14, 122.93, 118.44, 114.84, 114.79, 112.68, 112.55, 112.19, 109.74, 102.33, 102.11, 56.26, 55.73, 55.70, 55.66, 55.64, 45.16, 45.05, 35.77, 35.75, 35.39, 34.18, 28.11, 28.04, 27.92, 27.36, 27.10, 25.84, 21.92, 21.47, 15.23, 15.01, 14.92, 9.03.

### 3.6. General Procedure for the Synthesis of Compounds ***G4**–**G6***

1-Phenylamino-5-phenylimino-1,3-pentadiene hydrochloride (**5**) (500 mg, 1.76 mmol), 1-(5-carboxypentyl)-2,3,3,5-tetramethyl-3*H*-indol-1-ium bromide (**1a**) (1.30 g, 3.52 mmol), and triethylamine (1 mL) were dissolved in acetic anhydride, and the mixture was stirred at 60 °C for 1 h. Afterwards, the resulting solution was added to water and allowed to stand for 1 h followed by extraction with ethyl acetate. The organic layer was dried over Mg_2_SO_4_ and concentrated. The residue was purified by chromatography on silica gel to get **6a**. Next, DCC (86 mg, 0.419 mmol) and DMAP (5 mg, 0.04 mmol) were added to a solution of **6a** (300 mg, 0.419 mmol) in DCM in an ice-salt bath. After cooling to room temperature, INH (52 mg, 0.377 mmol) was added, and the mixture was stirred for 24 h. After filtration, the filtrate was evaporated in vacuo and the residue was purified by silica gel column to yield **G4** (60 mg, 17%). **G5** and **G6** were prepared by the same procedure as **G4**.

*2-(7-(1-(5-carboxypentyl)-1,3-dihydro-3,3,5-trimethyl-2H-indol-2-ylidene)-1,3,5-heptatrien-1-yl)-1-(6-(2-isonicotinoylhydrazinyl)-6-oxohexyl)-3,3,5-trimethyl-3H-indol-1-ium bromide* (**G4**). MS (ESI) *m*/*z*: 756.4 [M + H]^+^. ^1^H NMR (400 MHz, CD_3_OD) δ 8.73 (d, 2H), 7.87 (dd, 2H), 7.83 (d, 2H), 7.57–7.49 (br, 1H), 7.28 (s, 2H), 7.20 (d, 1H), 7.18 (d, 1H), 7.14 (t, 2H), 6.51 (t, 1H), 6.43 (t, 1H), 6.25 (d, 1H), 6.19 (d, 1H), 4.05 (dt, 4H), 2.39 (d, 6H), 2.36 (t, 2H), 2.25–2.18 (br, 2H), 1.88–1.82 (m, 2H), 1.81–1.76 (m, 4H), 1.68 (s, 2H), 1.66 (s, 6H), 1.65 (s, 6H), 1.57 (dt, 2H), 1.51–1.43 (m, 2H). ^13^C NMR (151 MHz, MeOD) δ 177.39, 174.89, 170.49, 166.76, 166.66, 159.54, 156.55, 151.24, 151.10, 142.14, 142.12, 141.94, 138.33, 130.83, 128.94, 126.46, 126.38, 125.30, 125.14, 123.16, 118.65, 114.71, 114.69, 112.48, 112.25, 112.23, 109.97, 104.82, 104.73, 66.87, 56.40, 55.53, 45.00, 44.97, 36.22, 35.46, 34.38, 28.38, 28.24, 27.79, 27.59, 27.24, 26.31, 26.06, 22.12, 15.44, 9.23.

*2-(7-(5-methoxyl-1-(5-carboxypentyl)-1,3-dihydro-3,3-dimethyl-2H-indol-2-ylidene)-1,3,5-heptatrien-1-yl)-5-methoxyl-1-(6-(2-isonicotinoylhydrazinyl)-6-oxohexyl)-3,3-dimethyl-3H-indol-1-ium bromide* (**G5**). MS (ESI) *m*/*z*: 788 [M + H]^+^. ^1^H NMR (400 MHz, CDCl_3_) δ 8.71 (d, 2H), 8.63–8.57 (br, 2H), 7.89–7.80 (br, 2H), 7.72 (d, 1H), 7.66 (d, 2H), 7.52–7.39 (m, 1H), 7.15–7.10 (m, 1H), 7.03–7.10 (br, 1H), 6.98 (d, 1H), 6.90 (d, 1H), 6.86 (s, 2H), 6.53–6.42 (br, 1H), 6.25 (d, 1H), 5.93 (d, 1H), 3.96–3.86 (br, 2H), 3.82 (s, 3H), 3.82 (s, 3H), 3.48 (q, 2H), 2.31 (d, 2H), 1.79–1.61 (br, 6H), 1.58 (s, 3H), 1.57 (s, 3H), 1.56–1.53 (m, 2H), 1.52–1.48 (br, 2H), 1.44–1.39 (br, 2H), 1.21 (t, 2H). ^13^C NMR (151 MHz, MeOD) δ 178.22, 174.91, 173.45, 172.13, 166.65, 159.65, 159.52, 156.50, 151.68, 151.21, 151.11, 144.33, 144.19, 141.94, 137.22, 130.81, 126.38, 123.15, 114.66, 114.65, 113.28, 112.65, 112.47, 110.96, 110.53, 109.98, 104.35, 104.32, 56.40, 50.42, 49.57, 44.96, 44.94, 36.93, 36.89, 34.38, 28.23, 28.08, 28.01, 27.58, 27.15, 26.56, 26.07, 24.62, 22.67.

*2-(7-(5-methoxyl-1-(5-carboxypentyl)-1,3-dihydro-5-ethyl-3-methyl-2H-indol-2-ylidene)-1,3,5-heptatrien-1-yl)-5-methoxyl-1-(6-(2-isonicotinoylhydrazinyl)-6-oxohexyl)-5-ethyl-3-methyl-3H-indol-1-ium bromide* (**G6**). MS (ESI) *m*/*z*: 816.6 [M + H]^+^. ^1^H NMR (400 MHz, CD_3_OD) δ 8.73 (s, 2H), 7.84 (d, 2H), 7.80 (d, 2H), 7.51–7.43 (br, 1H), 7.19 (t, 2H), 7.03 (t, 2H), 6.97–6.95 (m, 1H), 6.95–6.94 (m, 1H), 6.49 (t, 1H), 6.42 (t, 1H), 6.27 (d, 1H), 6.21 (d, 1H), 4.05 (dd, 4H), 3.84 (s, 3H), 3.83 (s, 3H), 2.36 (t, 2H), 2.32 (dd, 2H), 2.26–2.13 (br, 4H), 1.85–1.81 (m, 2H), 1.81–1.77 (m, 4H), 1.70–1.67 (br, 2H), 1.66 (s, 3H), 1.65 (s, 3H), 1.61–1.56 (br, 2H), 1.52–1.44 (br, 2H). ^13^C NMR (151 MHz, MeOD) δ 177.36, 174.89, 170.49, 166.76, 166.66, 159.54, 156.55, 151.24, 151.10, 151.05, 142.14, 142.12, 141.94, 138.33, 130.83, 128.94, 126.38, 125.30, 125.14, 123.16, 118.65, 114.71, 114.69, 112.48, 112.25, 112.23, 109.97, 104.82, 104.73, 66.87, 56.40, 55.53, 49.57, 45.00, 44.97, 36.22, 35.46, 34.38, 28.38, 28.24, 27.79, 27.59, 27.24, 26.31, 26.06, 22.12, 20.54, 15.44, 9.23.

### 3.7. General Procedure for the Synthesis of Compounds ***G7**–**G13***

1-Phenylamino-5-phenylimino-1,3-pentadiene hydrochloride (**5**) (500 mg, 1.76 mmol) and 4-(5-methoxy-2,3,3-trimethyl-3H-indol-1-ium-1-yl)butane-1-sulfonate (**2a**) were added to acetic anhydride with continuous stirring at 50 °C for 1 h. The mixture was poured into diethyl ether after cooling to room temperature. Then, precipitate (**A**) was filtered and used in further reaction without any purification. 

1-(5-Carboxypentyl)-2,3,3,5-tetramethyl-3*H*-indol-1-ium bromide (**1a**) (352 mg, 0.956 mmol) was added to a solution of **A** (500 mg, 0.956 mmol) in 25 mL pyridine. The resulting solution was stirred at 60 °C for 30 min. Diethyl ether was added in excess to the mixture to obtain the precipitate after cooling to room temperature. The resulting mixture was acidified with hydrochloric acid to get **6d**. Subsequently, DCC (92 mg, 0.445 mmol) and DMAP (5 mg, 0.04 mmol) were added to a solution of **6d** (300 mg, 0.445 mmol) in DCM in an ice-salt bath. After the mixture was cooled to room temperature, INH (76 mg, 0.557 mmol) was added slowly. The mixture was stirred for 24 h and then filtered. Filtrate was evaporated to dryness in vacuo and the residue was purified by silica gel to yield **G7** (80 mg, 23%). **G8**–**G13** were prepared by the same procedure as **G7**.

*3H-Indolium,2-(7-(5-methoxy-1-(4-sulfonatobutyl)-3,3-dimethyl-2H-indolin-2-ylidene)-1,3,5-heptatrien-1-yl)-1-(6-(2-isonicotinoylhydrazinyl)-6-oxohexyl)-1,3-dihydro-3,3,5-trimethyl-, inner salt* (**G7**). MS (ESI) *m*/*z*: 816.3 [M + Na]^+^. ^1^H NMR (400 MHz, CDCl_3_) δ 8.63 (s, 1H), 8.58–8.55 (br, 2H), 7.79–7.76 (br, 2H), 7.69 (s, 1H), 7.46–7.34 (m, 2H), 7.11–7.06 (br, 2H), 7.02 (d, 1H), 6.99 (s, 1H), 6.86 (d, 1H), 6.74 (s, 2H), 6.39–6.32 (br, 2H), 6.24–6.17 (br, 1H), 6.03–5.93 (br, 1H), 4.04–3.97 (br, 2H), 3.85–3.77 (br, 2H), 3.46 (s, 3H), 3.04–2.99 (m, 2H), 2.41–2.35 (br, 2H), 2.28 (s, 3H), 2.07–2.02 (br, 2H), 1.88–1.83 (br, 2H), 1.71–1.65 (m, 4H), 1.45 (s, 12H), 1.33 (d, 2H). ^13^C NMR (151 MHz, CD_3_OD) δ 174.66, 173.09, 171.11, 166.60, 166.40, 159.70, 150.91, 150.75, 144.27, 142.15, 141.74, 141.54, 136.74, 135.32, 129.82, 126.32, 124.88, 124.70, 123.75, 122.94, 122.60, 118.47, 114.63, 112.88, 112.30, 110.87, 109.73, 103.44, 56.23, 55.55, 51.58, 50.49, 49.61, 44.92, 44.44, 43.57, 34.24, 27.95, 27.77, 27.71, 27.22, 27.00, 25.92, 23.31, 21.06.

*3H-Indolium,2-(7-(5-methoxy-1-(4-sulfonatobutyl)-3,3-dimethyl-2H-indolin-2-ylidene)-1,3,5-heptatrien-1-yl)-5-methoxy-1-(6-(2-isonicotinoylhydrazinyl)-6-oxohexyl)-1,3-dihydro-3,3-dimethyl-, inner salt* (**G8**). MS (ESI) *m*/*z*: 832.3 [M + Na]^+^. ^1^H NMR (400 MHz, CDCl_3_) δ 8.46–8.35 (br, 2H), 7.72–7.62 (br, 2H), 7.47 (d, 2H), 6.99–6.90 (br, 2H), 6.87–6.83 (br, 1H), 6.82–6.76 (br, 2H), 6.75–6.70 (br, 2H), 6.64–6.60 (br, 1H), 6.60–6.55 (br, 1H), 6.32–6.22 (br, 2H), 6.05–5.98 (br, 1H), 5.97–5.89 (br, 1H), 3.83–3.75 (br, 2H), 3.71 (s, 3H), 3.52 (s, 3H), 2.92–2.82 (br, 2H), 2.33–2.19 (br, 2H), 1.93–1.82 (br, 2H), 1.57–1.43 (m, 6H), 1.39 (s, 6H), 1.34 (s, 6H), 1.29–1.25 (m, 4H). ^13^C NMR (151 MHz, CD_3_OD) δ 174.84, 172.15, 171.92, 166.84, 166.59, 159.52, 159.43, 151.12, 150.96, 144.19, 144.08, 141.91, 137.19, 137.16, 128.85, 126.48, 125.17, 124.99, 123.16, 122.84, 118.69, 114.67, 114.63, 112.55, 112.48, 109.95, 104.52, 104.32, 66.85, 56.41, 55.61, 51.83, 50.42, 50.34, 45.01, 44.82, 43.64, 34.41, 28.10, 28.03, 27.33, 27.15, 26.08, 23.51, 23.42.

*3H-Indolium,2-(7-(5-methoxy-1-(4-sulfonatobutyl)-3,3-dimethyl-2H-indolin-2-ylidene)-1,3,5-heptatrien-1-yl)-1-(6-(2-isonicotinoylhydrazinyl)-6-oxohexyl)-1,3-dihydro-3,3-dimethyl-, inner salt* (**G9**). MS (ESI) *m*/*z*: 780.3 [M + H]^+^. ^1^H NMR (400 MHz, CD_3_OD) δ 8.68 (d, 2H), 7.95–7.89 (m, 1H), 7.86 (d, 2H), 7.79 (t, 1H), 7.70–7.64 (m, 1H), 7.57–7.46 (m, 1H), 7.39 (d, 1H), 7.36–7.31 (m, 2H), 7.26–7.23 (m, 1H), 7.17–7.14 (m, 1H), 6.99 (dd, 1H), 6.54–6.45 (m, 2H), 6.40 (d, 1H), 6.12 (d, 1H), 4.21–4.15 (m, 2H), 4.02–3.98 (m, 2H), 3.85 (s, 3H), 2.89 (t, 2H), 2.35 (t, 2H), 2.01–1.94 (m, 2H), 1.96–1.93 (m, 2H), 1.83–1.77 (m, 4H), 1.69 (s, 6H), 1.66 (s, 6H), 1.58–1.55 (br, 2H). ^13^C NMR (151 MHz, MeOD) δ 174.90, 174.49, 170.13, 166.66, 160.34, 151.12, 150.96, 150.25, 144.81, 144.11, 141.86, 136.69, 129.54, 128.91, 126.61, 125.21, 124.98, 124.78, 123.16, 122.82, 118.64, 115.00, 113.64, 112.52, 110.86, 109.90, 106.59, 102.93, 66.87, 56.44, 54.82, 51.71, 51.10, 45.40, 44.34, 34.42, 28.25, 27.83, 27.78, 27.51, 27.22, 26.13, 23.48, 15.44.

*2-(7-(5-Methoxy-1-butyl-3,3,5-trimethyl-2H-indolin-2-ylidene)-1,3,5-heptatrien-1-yl)-1-(6-(2-isonicotinoylhydrazinyl)-4-oxobutyl)-1,3-dihydro-3,3-dimethyl-3H-indolium bromide* (**G10**). MS (ESI) *m*/*z*: 686.4 [M − Br]^+^. ^1^H NMR (600 MHz, CDCl_3_) δ 8.61 (d, 2H), 7.84 (d, 2H), 7.67–7.61 (m, 2H), 7.60–7.53 (m, 1H), 7.34–7.28 (br, 1H), 7.13 (s, 2H), 7.10 (d, 1H), 7.06 (s, 1H), 6.91 (d, 1H), 6.88 (s, 1H), 6.83 (d, 1H), 6.62–6.54 (br, 1H), 6.35 (dd, 2H), 5.89 (d, 1H), 4.19–4.14 (br, 2H), 3.91–3.84 (m, 2H), 3.82 (s, 3H), 3.14 (td, 2H), 2.70–2.64 (br, 2H), 2.34 (s, 3H), 2.10–2.04 (m, 2H), 1.75–1.69 (m, 2H), 1.60 (s, 6H), 1.57 (s, 6H), 0.97 (t, 3H). ^13^C NMR (151 MHz, MeOD) δ 173.68, 171.84, 170.86, 165.09, 160.04, 156.87, 152.97, 151.03, 150.54, 149.88, 144.64, 144.46, 142.21, 141.67, 136.89, 135.35, 130.01, 126.79, 126.65, 123.94, 123.23, 122.95, 114.87, 113.15, 110.87, 109.91, 105.64, 103.38, 56.42, 50.86, 49.71, 47.36, 45.23, 43.87, 31.70, 30.80, 28.18, 27.84, 27.33, 23.97, 21.25, 21.12, 14.23.

*2-(7-(5-Methoxy-1-butyl-3,3,5-trimethyl-2H-indolin-2-ylidene)-1,3,5-heptatrien-1-yl)-1-(6-(2-isonicotinoylhydrazinyl)-4-oxohexyl)-1,3-dihydro-3,3-dimethyl-3H-indolium bromide* (**G11**). MS (ESI) *m*/*z*: 714.5 [M − Br]^+^. ^1^H NMR (400 MHz, CDCl_3_) δ 8.62 (s, 2H), 7.67 (s, 4H), 7.43–7.36 (br, 1H), 7.11 (d, 1H), 7.08 (s, 1H), 6.95 (t, 2H), 6.89 (d, 1H), 6.86 (dd, 1H), 6.55–6.48 (m, 1H), 6.40 (t, 1H), 6.06 (d, 1H), 5.98 (d, 1H), 3.90 (s, 4H), 3.83 (s, 3H), 2.34 (s, 3H), 1.63 (s, 6H), 1.60 (s, 6H), 1.54–1.50 (m, 2H), 1.43 (dq, 2H), 1.32 (d, 2H), 0.96 (t, 3H). ^13^C NMR (151 MHz, CD_3_OD) δ 173.32, 172.08, 171.28, 164.43, 159.87, 156.96, 152.77, 151.27, 150.28, 149.73, 145.36, 144.53, 142.33, 141.70, 136.93, 135.45, 130.00, 126.65, 126.43, 123.92, 123.21, 122.90, 114.80, 112.98, 111.00, 109.89, 105.29, 103.47, 66.86, 56.42, 50.74, 49.78, 45.12, 44.55, 35.21, 30.74, 28.15, 28.02, 27.89, 27.30, 26.52, 21.26, 21.10, 15.43, 14.24.

*2-(7-(1-Butyl-3,3,5-trimethyl-2H-indolin-2-ylidene)-1,3,5-heptatrien-1-yl)-5-bromo-1-(6-(2-isonicotinoylhydrazinyl)-4-oxohexyl)-1,3-dihydro-3,3-dimethyl-3H-indolium bromide* (**G12**). MS (ESI) *m*/*z*: 778.3 [M − Br]^+^. ^1^H NMR (600 MHz, CDCl_3_) δ 8.60 (s, 2H), 7.75 (t, 1H), 7.70–7.65 (br, 2H), 7.57 (t, 1H), 7.41 (t, 1H), 7.36 (d, 1H), 7.29 (s, 1H), 7.07 (d, 1H), 6.92 (s, 1H), 6.90 (d, 1H), 6.85 (d, 1H), 6.46 (dd, 2H), 6.17 (d, 1H), 5.90 (d, 1H), 4.03–3.98 (br, 2H), 3.84 (s, 3H), 3.80–3.76 (br, 2H), 2.36–2.29 (br, 2H), 1.78–1.75 (m, 2H), 1.72–1.67 (br, 4H), 1.64 (s, 6H), 1.57 (s, 6H), 1.50–1.46 (br, 2H),1.45– 1.42 (m, 2H), 0.96 (t, 3H). ^13^C NMR (151 MHz, CD_3_OD) δ 175.73, 174.00, 167.39, 165.87, 160.66, 156.27, 154.33, 150.69, 150.18, 148.38, 145.15, 143.60, 143.53, 142.60, 136.21, 132.08, 127.55, 126.77, 126.22, 122.96, 122.74, 116.16, 115.09, 114.04, 111.64, 109.59, 107.76, 102.00, 66.68, 56.29, 51.37, 47.13, 45.62, 34.43, 30.84, 28.05, 27.42, 27.39, 27.16, 27.11, 27.03, 26.05, 20.87, 13.99.

*1H-Benz[e]indolium,2-(7-(3,3-dimethyl-1-(4-sulfonatobutyl)benz(e)indolin-2-ylidene)hepta-1,3,5-trien-1-yl)-1-(6-(2-isonicotinoylhydrazinyl)-6-oxohexyl)-1,3-dihydro-3,3-dimethyl-, inner salt* (**G13**). MS (ESI) *m*/*z*: 872.5 [M + H]^+^. ^1^H NMR (400 MHz, CDCl_3_) δ 8.65 (d, 1H), 8.55 (s, 4H), 7.99–7.91 (m, 2H), 7.84–7.79 (br, 4H), 7.74 (s, 2H), 7.68 (d, 1H), 7.60–7.56 (br, 1H), 7.53–7.46 (m, 2H), 7.42–7.37 (br, 1H), 7.35 (d, 2H), 7.31–7.29 (m, 1H), 7.22–7.10 (br, 1H), 6.61–6.52 (br, 1H), 6.50–6.43 (br, 1H), 6.37–6.28 (br, 1H), 6.26–6.17 (br, 1H), 4.17–4.07 (br, 2H), 4.07–3.95 (br, 2H), 3.08–3.00 (br, 2H), 2.47–2.31 (br, 2H), 2.07 (s, 4H), 1.98–1.89 (br, 2H), 1.78 (s, 6H), 1.75 (s, 6H), 1.58–1.50 (m, 2H). ^13^C NMR (151 MHz, MeOD) δ 174.85, 174.48, 174.24, 166.89, 166.59, 151.07, 151.00, 150.94, 144.53, 142.01, 141.11, 134.80, 134.64, 133.26, 133.23, 131.69, 131.05, 130.82, 129.44, 128.85, 128.66, 128.62, 127.17, 125.92, 125.84, 125.14, 124.94, 123.31, 123.27, 123.15, 122.82, 118.67, 112.50, 112.08, 112.05, 104.47, 66.88, 55.63, 54.82, 52.13, 51.82, 45.00, 43.65, 34.41, 28.36, 27.61, 27.12, 26.10, 23.79, 23.52, 15.43.

## 4. Conclusions

In this study, we developed a series of novel near-infrared heptamethine dye–INH conjugates. Evaluation of IC_50_ values indicated that nearly all conjugates showed moderate antitumor efficacy. Among them, conjugates **G10** and **G11** exhibited strong cytotoxicity comparable to doxorubicin in inhibiting PC-3 cell growth while INH showed negligible in vitro antitumor activity. Furthermore, MAOA inhibitory activities of **G10**, **G11**, and **G12** and molecular docking analysis were consistent with the cell viability results, suggesting that the enhanced antitumor activity of these conjugates may be attributed to the preferential accumulation in cancer cells mediated by OATPs and the increased MAOA inhibitory activity derived from INH N-acylation. All these above results suggested that a combination of the MAOA inhibitor INH and tumor-specific targeting heptamethine cyanine dyes may provide a promising tool for the treatment of advanced prostate cancer.

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
