# Peer review of "Design, Synthesis, and Evaluation of Monoamine Oxidase A Inhibitors–Indocyanine Dyes Conjugates as Targeted Antitumor Agents"

_molecules, 2019, doi:10.3390/molecules24071400_

Reviewer 1 Report

The manuscript has significantly improved and can be accepted after minor revision

The Authors stated "As illustrated in Scheme 2, symmetric heptamethine cyanine dye intermediates 7 were synthesized.. [....], bur intermediates 7 are  not reported in the Scheme

It is not clear if cytotoxicity and MAO inhibition assays (paragraphs 3.3 and 3.4)been performed on time or if at least two replicates have been performed.

In Paragraph 3.4 the Authors referred to "various concentrations", please specify, Moreover, they stated that they used a specific kit, please insert the name and the appropriate reference.

Paragraph 3.7: 

-please replace "in pyridine of 25 mL" with "25mL pyridine", and

- the sentence "The mixture was added with excessive diethyl ether" with "diethyl ether was added in excess to the mixture":

Author Response

Point 1: The Authors stated "As illustrated in Scheme 2, symmetric heptamethine cyanine dye intermediates 7 were synthesized.. [....], bur intermediates 7 are  not reported in the Scheme

Response 1: In the sentence “As illustrated in Scheme 2, symmetric heptamethine cyanine dye intermediates 7 were synthesized”, intermediates 7 have been recorrected as intermediates 6.

Point 2: It is not clear if cytotoxicity and MAO inhibition assays (paragraphs 3.3 and 3.4)been performed on time or if at least two replicates have been performed.

Response 2: Cytotoxicity and MAOA inhibition assays (paragraphs 3.3 and 3.4) have been repeated.

Point 3: In Paragraph 3.4 the Authors referred to "various concentrations", please specify, Moreover, they stated that they used a specific kit, please insert the name and the appropriate reference.

Response 3: In paragraph 3.4 we have modified as “with various concentrations (10 pM, 1 nM, 100 nM, 10 μM)”. The specific kit is Cell MAOA Assay Kit (GenMed Scientifics Inc. U.S.A, Lot No: 1-201838-10) which was never reported in references.

Point 4: Paragraph 3.7:

-please replace "in pyridine of 25 mL" with "25mL pyridine", and

- the sentence "The mixture was added with excessive diethyl ether" with "diethyl ether was added in excess to the mixture":

Response 4: We have modified the related parts to “To a solution of A (500 mg, 0.956 mmol) in 25 mL pyridine was added… ” and “Diethyl ether was added in excess to the mixture to obtain the precipitate after cooling to room temperature”.

Reviewer 2 Report

All queries are well addressed. I am pleased to accept the current form of this manuscript for publication in Molecules

Author Response

Thanks for your comments.

This manuscript is a resubmission of an earlier submission. The following is a list of the peer review reports and author responses from that submission.

Round  1

Reviewer 1 Report

The manuscript of Guo et al regards the synthesis and evaluation of Monoamine oxidase A (MAOA) iinhibitors conjugates for targeting tumor. They  synthesized a series of compounds and evaluated their in vitro cytotoxicity against PC-3 cells.  Three compounds showed a strong inhibitory effect against PC-3 cells. The Authors affirm that  the docking studies were consistent with the cytotoxicity results. 

In my opinion molecular modelling studies are not sufficient for stating that the cytotoxicicty is  MAOA mediated. I suggest to improve the discussion regarding this point.

Moreover, since the function of MAOA in cancer is still under discussion the following sentence should be repharsed "These molecular docking results, suggested that a combination of  antidepressant drugs against MAOA with high tumor targeting heptamethine cyanine dyes may  prove to be highly promising tools for the treatment of advanced prostate cancer."

The references are not up to date. Just one reference is dated 2017.

See:

Lv, Q., Wang, D., Yang, Z., Yang, J., Zhang, R., Yang, X., Wang, M., Wang, Y. Repurposing antitubercular agent isoniazid for treatment of prostate cancer (2018) Biomaterials science, 7 (1), pp. 296-306.  https://www.scopus.com/inward/record.uri?eid=2-s2.0- Liao, C.-P., Lin, T.-P., Li, P.-C., Geary, L.A., Chen, K., Vaikari, V.P., Wu, J.B., Lin, C.-H., Gross, M.E., Shih, J.C. Loss of MAOA in epithelia inhibits adenocarcinoma development, cell proliferation and cancer stem cells in prostate (2018) Oncogene, 37 (38), pp. 5175-5190. Cited 1 time. Kim, Y., Pierce, C.M., Robinson, L.A. Impact of viral presence in tumor on gene expression in non-small cell lung cancer(2 018) BMC Cancer, 18 (1), art. no. 843, .  Lv, Q., Yang, X., Wang, M., Yang, J., Qin, Z., Kan, Q., Zhang, H., Wang, Y., Wang, D., He, Z. Mitochondria-targeted prostate cancer therapy using a near-infrared fluorescence dye–monoamine oxidase A inhibitor conjugate (2018) Journal of Controlled Release, 279, pp. 234-242.  Liu, F., Hu, L., Ma, Y., Huang, B., Xiu, Z., Zhang, P., Zhou, K., Tang, X. Increased expression of monoamine oxidase A isassociated with epithelial to mesenchymal transition andclinicopathological features in non-small cell lung cancer (2018) Oncology Letters, 15 (3), pp. 3245-3251.  Sun, W.Y., Choi, J., Cha, Y.J., Koo, J.S. Evaluation of the expression of amine oxidase proteins in breast cancer (2017) International Journal of Molecular Sciences, 18 (12), art. no. 2775,  Li, P.C., Siddiqi, I.N., Mottok, A., Loo, E.Y., Wu, C.H., Cozen, W., Steidl, C., Shih, J.C. Monoamine oxidase A is highly expressed in classical Hodgkin lymphoma (2017) Journal of Pathology, 243 (2), pp. 220-229.

Lastly, the manuscript should be revised by a native english speaker. See as an example this sentence "we designed and fabricated a series of novel conjugates"

Author Response

I really appreciate your comments for the manuscript. According to your advice, we amended the relevant part in manuscript. Some of your questions were answered below.

Reviewer 2 Report

In the manuscript entitled “Design, synthesis, and evaluation of potential monoamine oxidase A inhibitors-indocyanine dyes conjugates as targeted antitumor agents” Xiao-Guang Yang and co-workers have synthesized some conjugates of heptamethine carbocyanine dyes and isoniazide, and investigated their cytotoxic effects against PC-3 cell lines. Three compounds (G10, G11, and G12) exhibited potent anti-tumor activities against PC-3 cells compared to Isoniazid and DOX. I found this manuscript of limited interest and has some serious concerns:

1)      Why authors have chosen Isoniazid as standard drug. Isoniazid is an antibiotic used for the treatment of tuberculosis and has no any cytotoxic effect on PC-3 cells?

2)      Why authors have chosen Isoniazid scaffold to design these conjugates? Is there any specific reason for this? Write more about rational design of these conjugates.

3)      The biological studies are not complete. The authors have just evaluated cytotoxic effects of these compounds against PC-3 cells. They should also find their effects against Monoamine oxidase A (MAOA) and must compare with Clorgyline.

Author Response

I really appreciate your comments for the manuscript. According to your advice, we amended the relevant part in manuscript. Some of your questions were answered below.

Molecules EISSN 1420-3049 Published by MDPI AG, Basel, Switzerland RSS E-Mail Table of Contents Alert
Back to Top